# CFD Prediction of Wind Loads on FPSO and Shuttle Tankers during Side-by-Side Offloading

**Jung-Hee Yoo [1], Patrick Schrijvers [2], Arjen Koop [3] and Jong-Chun Park [4,*]**

1   Marine Leisure & Safety Research Center, Research Institute of Medium & Small Shipbuilding, Busan 46757, Korea; jhyoo@rims.re.kr
2   Offshore, Maritime Research Institute Netherlands (MARIN), Haagsteeg 2, 6708 PM Wageningen, The Netherlands; p.schrijvers@marin.nl
3   Research & Development, Maritime Research Institute Netherlands (MARIN), Haagsteeg 2, 6708 PM Wageningen, The Netherlands; a.koop@marin.nl
4   Naval Architecture & Ocean Engineering, Pusan National University, Busan 46241, Korea
*   Correspondence: jcpark@pnu.edu; Tel.: +82-51-510-2480

**Abstract:** This study entailed the estimation of wind loads performed using computational fluid dynamics (CFD) simulations for four typical offshore vessels and for a Floating Production, Storage, and Offloading (FPSO) and shuttle tanker in a side-by-side configuration on offloading. For all vessels, under the wind heading condition, four meshes were used to carry out the verification and validation (V&V) study to check the numerical uncertainty. The CFD simulation results for the aerodynamic coefficients were compared with wind tunnel tests from the Offloading Operability 2 JIP. All CFD simulation results show generally good agreement with the experimental data, and the overall trend of the coefficients are well captured. In addition, the effect on the gap sizes between the FPSO and shuttle tanker in the range of 4–30 m was examined. On this basis, the shielding effect was analyzed according to the size of the gap between the two ships.

**Keywords:** wind load; Computational Fluid Dynamics (CFD); FPSO; shuttle tanker; membrane-type LNGC; moss-type LNGC; atmospheric boundary layer; side-by-side offloading configuration

## 1. Introduction

Wind loads on operating vessels and offshore structures can significantly affect the ship motions, maneuvering situations, and mooring of offshore constructions at sea. The performance and safety of floating vessels depend on an accurate prediction of wind loads at the design stage. The impact of wind loads has become increasingly important as vessels have become larger and operating speeds have increased [1,2]. Wind loads can reach approximately 20% of the total load on the vessel and lead to an increase in the required horsepower, especially in rough weather [3,4].

To date, wind tunnel tests have been commonly used to estimate wind loads. A wide range of wind tunnel tests for various types of ships were carried out to investigate the air resistance and wake generated by superstructures [5–10] and to determine wind forces and wind load coefficients [11–17]. However, they have some limitations such as scale effect, blockage effect due to the limited tunnel size, and difficulties in modeling the atmospheric boundary layer profile. Recently, computational fluid dynamics (CFD) simulations have been accompanied by wind tunnel tests in various engineering fields [18–22]. CFD modeling can be an insightful analysis tool with a similar accuracy compared to experimental approaches. One of the strongest advantages of numerical simulation is that it can easily provide a detailed view of the full range of flow characteristics around target objects.

There are several important modeling factors for CFD simulations of wind loads, such as geometry modeling, correct representation of the atmospheric boundary velocity profile, and grid dependency. Koop et al. (2010) [23] reported that the simulated drag



coefficients differ by approximately 20% between two cases using the geometry of the Floating Production, Storage, and Offloading (FPSO) used in the experiment and the simplified geometry. In addition, the simulated drag coefficients using different inlet velocity profiles can vary by as much as 60% [24,25]. In Koop et al. (2010) [23], on the other hand, the wind load coefficients were determined for a shuttle tanker, FPSO, and tandem-offloading configuration with the shuttle tanker located in the wake of an FPSO. They concluded that the wind loads on single vessels could be obtained with reasonable accuracy compared to the experimental data; however, for the loads on the shielded vessel, more research is required. The CFD capabilities for estimating wind loads have increased since 2010, and it is expected that the reasonable accuracy achieved by [23] could be improved more drastically in the near future.

In this study, first, the CFD simulations for the estimation of wind loads for a single vessel of the FPSO, shuttle tanker, membrane-type, and moss-type Liquefied natural gas (LNG) carriers (LNGC) will be performed. In addition, a verification study was conducted to determine the numerical uncertainty with four different grids for each vessel and heading angle. In the validation process, the numerical verification and validation (V&V) tool suggested by [26] was employed, which was based on the V&V work in [27].

Furthermore, the CFD prediction of wind loads during a side-by-side configuration is also investigated with different gap sizes between two vessels ranging from 4–30 m. The simulation results were compared with those of the wind tunnel test that was carried out within the offshore operability 2 (OO2) JIP. Therefore, the main goal of this study is to estimate the wind loads on vessels and validate the CFD simulation results through a comparison with experimental data. More specifically, the objectives are as follows:

- to obtain wind load coefficients on single vessels of different shapes;
- to assess the current status of wind load simulations for single vessels in terms of accuracy;
- to determine the numerical accuracy in side-by-side configuration; and
- to investigate the effect of gap size on shielding coefficients in CFD.

## 2. Numerical Solver and Experimental Data

### 2.1. CFD Code: ReFRESCO

In this study, all simulations were performed using ReFRESCO (Available online: https://www.refresco.org (accessed on 8 May 2022)), which is a community-based open-usage/open-source CFD code for maritime applications. It can solve multiphase (unsteady) incompressible viscous flows using the Reynolds-averaged Navier-Stokes (RANS) equations, complemented with turbulence models, cavitation models, and volume-fraction transport equations for different phases [28]. In this study, however, the following continuity and RANS equations were introduced as the governing equations to simply solve compressible viscous fluids:

$$\frac{\partial U_i}{\partial x_i} = 0 \tag{1}$$

$$\frac{\partial U_i}{\partial t} + U_j \frac{\partial U_i}{\partial x_j} = -\frac{1}{\rho}\frac{\partial p}{\partial x_i} + \frac{\partial}{\partial x_i}\left(\nu \frac{\partial U_i}{\partial x_j} - \overline{u_i u_j}\right) + g_i, \tag{2}$$

where $U_i$ is the velocity, $\rho$ is the density of fluid, $t$ is time, $x_i$ is the position of fluid, $p$ is the pressure, $\nu$ is kinmatic viscosity, $\overline{u_i u_j}$ is the turbulence shear stress and $g_i$ is the gravity.

The governing equations are discretized using a finite-volume approach with cell-centered collocated variables in strong-conservation form, and a pressure-correction equation based on the SIMPLE algorithm is used to ensure mass conservation. At each implicit time step, the nonlinear system for velocity and pressure is linearized using Picard's method, and either a segregated or coupled approach is used [29].

The implementation is face-based, which permits grids with elements consisting of an arbitrary number of faces (hexahedrons tetrahedrons, prisms, pyramids, etc.), as well as h-refinement (hanging nodes). State-of-the-art CFD features such as moving, sliding,

and deforming grids, as well as automatic grid adaptation (refinement and/or coarsening) are available. Coupling with rigid-body (rigid-body 6-DOF) and flexible-body structural equations of motion (Fluid-Structure-Interaction, FSI) is also possible. For turbulence modeling, both RANS/URANS and Scale-Resolving Simulation (SRS) models, such as SAS, DDES/IDDES, XLES, PANS, and LES approaches, can be used. The code is parallelized using MPI and subdomain decomposition and runs on Linux workstations and HPC clusters. Couplings with propeller models (RANS-BEM coupling), fast-time simulation tools (XMF), and wave generation potential flow codes (OceanWave3D, SWASH) are possible.

The ReFRESCO is currently being developed and verified, and its several applications are being validated at MARIN (Wageningen, The Netherlands) in collaboration with various universities.

### 2.2. Atmospheric Boundary Layer Modeling

The atmospheric boundary layer (ABL) plays an important role in the wind load on offshore constructions [30]. Offshore structures are mainly located in this boundary layer and are affected by the variation in velocity depending on the height of the structure.

The wind profile introduced in this study and the wind tunnel test can be described as:

$$U(z) = CU_{ref}\left(\frac{z}{z_{ref}}\right)^{\alpha}, \tag{3}$$

where $U_{ref}$ is the reference velocity, typically at $Z_{ref} = 10$ m height, $C = 1.0$ is a constant, and $\alpha = 0.12$ is the power exponent describing the height dependence, which was obtained by tuning the theoretical profile to the wind tunnel result [23].

Figure 1 presents a comparison of the velocity profile with the target profile and the differences in velocity versus height, in which the target velocity profile was used in the wind tunnel test and the other profiles were obtained from the CFD simulation at different locations in the computational domain. To generate the wind profile, as established through sensitivity tests, the surface roughness was set to 2.189, and the eddy viscosity ($\nu_T/\nu$) and $y^+$ were set to 10 and 1, respectively. It can be seen that the differences between CFD and the target profile at the target location where a model is to be positioned are less than 1.3%.

### 2.3. Uncertainty of Wind Tunnel Experiments

During the Offshore Operability 2 JIP, a series of wind tunnel tests were conducted in a DNW-LST wind tunnel [31]. Figure 2 shows the models used in the test. The scale of the models was set at 1:250. Four repeated tests were conducted for the moss-type LNG carrier, which gives an estimate of the experimental uncertainty. For example, the measured $C_x$ according to the wind headings, as indicated in Figure 3a, the standard deviation for each heading cannot be directly determined because the loads are not identical between repeated tests. Therefore, a polynomial trend line was fitted through the measured data, as shown in Figure 3b. Based on the fitted polynomial function, the mean coefficient and standard deviation for a specific heading were determined, as presented in Table 1. The numerical uncertainty obtained in this manner is the highest with a heading of approximately 90° for $C_x$, with a maximum uncertainty of up to 54%, but this may not be the most relevant coefficient for that heading. The experimental uncertainty contains more components than the repeatability, which are not estimated here.

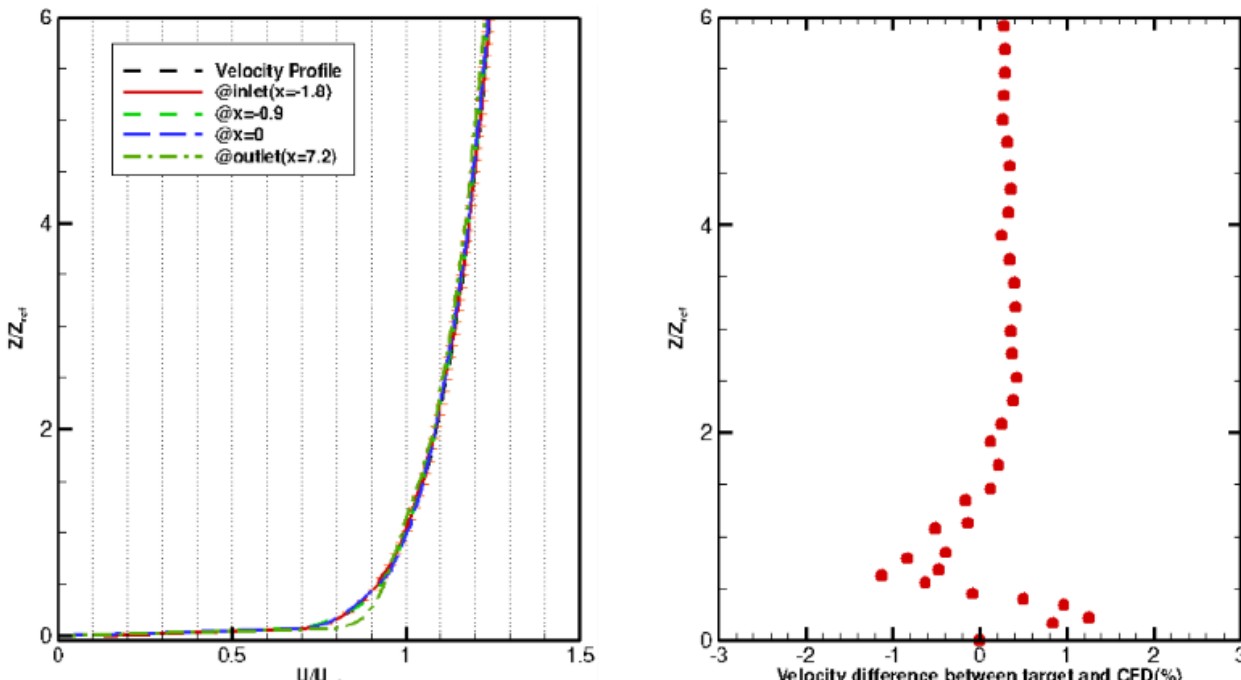

**Figure 1.** Comparison of velocity profiles employed in the wind tunnel test (dashed black line) and CFD (colored lines) in the (**left panel**), and the difference between the target profile and CFD profile at the center of the domain in the (**right panel**).

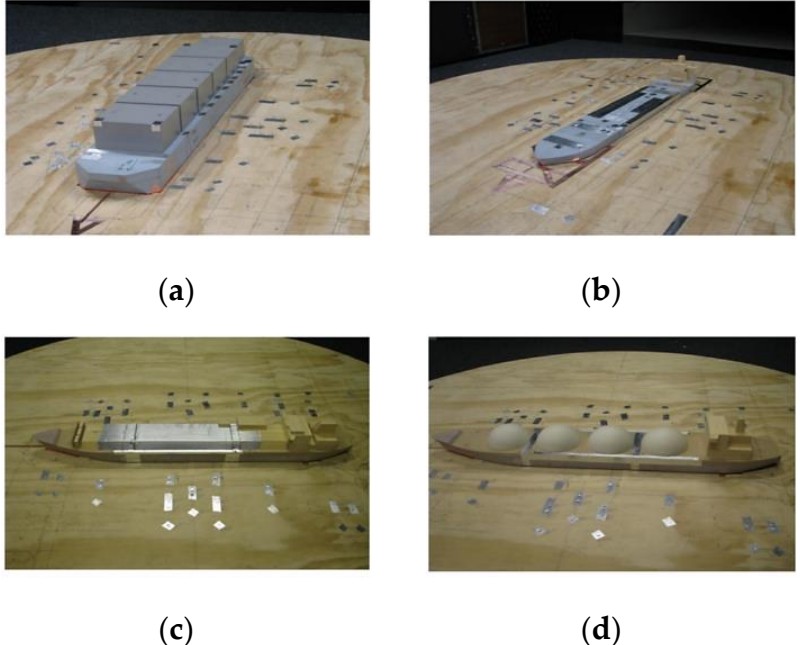

**Figure 2.** Vessel models for wind tunnel tests. (**a**) FPSO; (**b**) shuttle tanker; (**c**) membrane-type LNGC; and (**d**) moss-type LNGC.

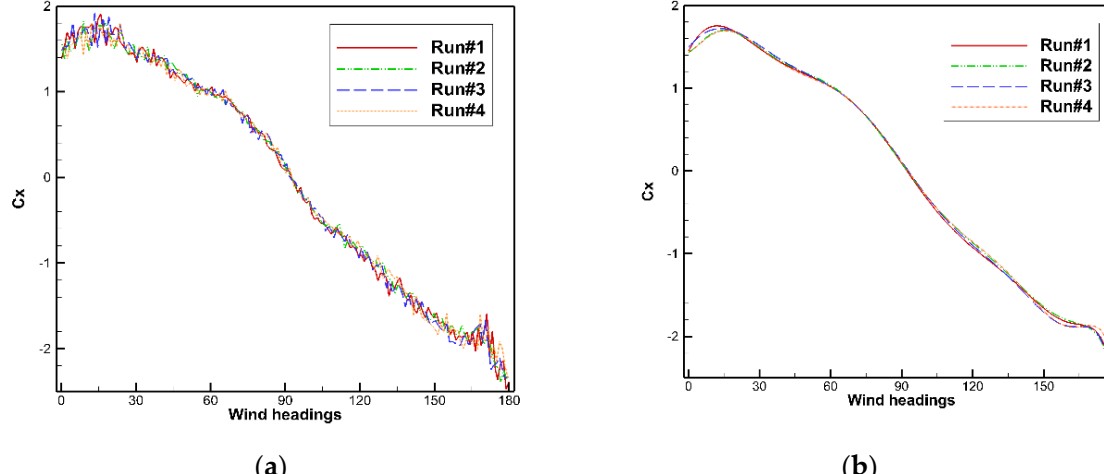

(**a**)                                        (**b**)

**Figure 3.** Comparison of four wind tunnel tests of moss-type LNGC. Schemes follow the same formatting. (**a**) Measured x-force coefficients; and (**b**) trend lines.

**Table 1.** Experimental uncertainty analysis for moss-type LNGC.

| Headings (°) | Mean (m/s) | Standard Deviation (m/s) | Uncertainty (%) |
|:---:|:---:|:---:|:---:|
| 0 | 1.45 | 0.025 | 1.7 |
| 45 | 1.22 | 0.014 | 1.2 |
| 90 | 0.10 | 0.009 | 9.6 |
| 135 | −1.25 | 0.018 | −1.5 |
| 180 | −2.62 | 0.149 | −5.7 |

## 3. Wind Loads on Single Vessels

### 3.1. Numerical Setup

For the CFD simulations, the wind tunnel geometry for each vessel was set to a scale of 1:250.

As shown in Figure 4, a cylindrical domain was employed such that one mesh can be used for all headings. The radius of the domain was set to four times the ship length ($= 4L_{ref}$), which means that the domain size will vary for each vessel. The top boundary is at a height of 2.25 m and is identical to that of the wind tunnel section. In other words, it can be said that no blockage correction was applied in the computational domain of the present simulation because the blockage effect was low (<0.5% for each vessel).

For each vessel, four grids with different grid numbers were generated using Hexpress 5.2, a commercial software used for grid generation. Figure 5 shows an overview of the coarsest mesh for each vessel. The number of cells used for each mesh is listed in Table 2. Cells (or control volumes) were locally refined on the vessel surfaces and in the wake of each vessel.

The average $y^+$ is below 0.5 on the vessel surfaces, and no wall functions were adopted. Considering the steady state, the momentum equation is solved using the higher-order QUICK scheme, and the SST k–ω turbulence model is introduced [32].

At the side and outflow boundaries, a constant static pressure was specified, and a zero normal gradient condition was imposed for the velocity. At the top boundary of the numerical domain, a free-slip wall condition was imposed, and the vessel surfaces were treated as a no-slip boundary.

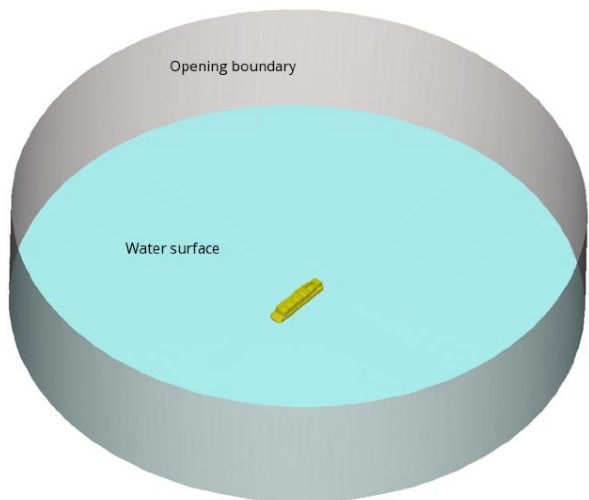

**Figure 4.** Computational domain for wind loads on FPSO.

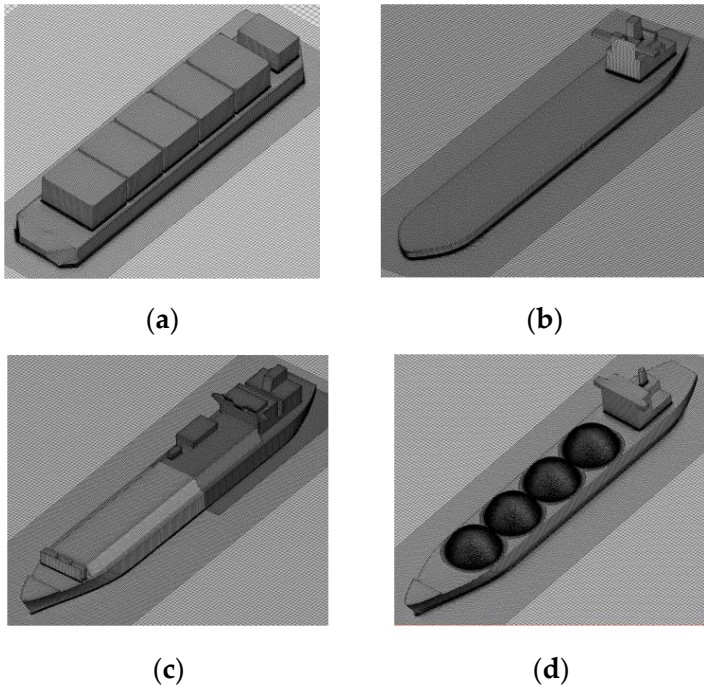

**Figure 5.** Computational grid system for wind load on offshore vessels. (**a**) FPSO; (**b**) Shuttle tanker; (**c**) Membrane-type LNGC; (**d**) Moss-type LNGC.

**Table 2.** Computational meshes used for CFD calculations.

| Vessels | Number of Cells | | | |
| --- | --- | --- | --- | --- |
| | Case#1 | Case#2 | Case#3 | Case#4 |
| FPSO | 22.5M | 10M | 6.7M | 5M |
| Shuttle tanker | 23M | 9M | 6M | 4.1M |
| Membrane LNGC | 35M | 15M | 9.2M | 6M |
| Moss LNGC | 37M | 18M | 12M | 9.5M |

To check the convergence of the numerical simulations, the residuals for the shuttle tanker and for the aerodynamic forces and moments on the membrane-type LNGC are plotted in Figures 6 and 7, respectively. For all vessels, 5000 iterations were conducted, and residuals stagnated over 1500 iterations. Figure 6 shows that using a finer mesh allows a slightly further reduction in the residuals. It is ensured that numerical convergence is sufficient, with a reduction in the residuals $L_2$-norm by three or four orders of magnitude. Additionally, when checking the forces and moments acting on the vessel along the iteration history, as shown in Figure 7, it can be confirmed that they converge at approximately 2500 iterations. The reported coefficients are the average of 3000 to 5000 iterations. To check the independence of mesh, some samples of the results for the FPSO are shown in Figure 8. In most cases, there is no significant difference between the results of cases #3 and #4.

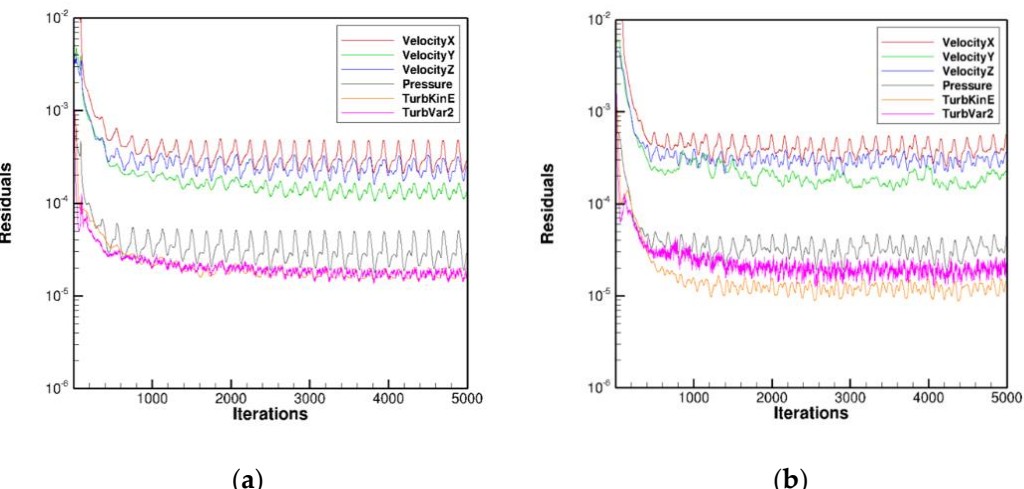

(**a**)                                 (**b**)

**Figure 6.** Residuals $L_2$-norm of the shuttle tanker simulations, $\alpha = 0°$. (**a**) finest mesh; (**b**) coarsest mesh.

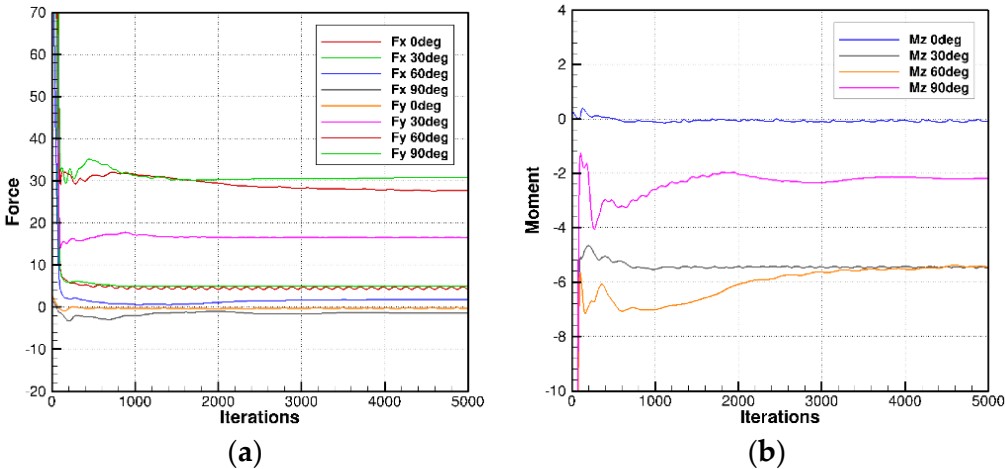

(**a**)                                 (**b**)

**Figure 7.** Convergence of forces and moments for the membrane-type LNGC, $\alpha = 0°$, $30°$, $60°$, and $90°$. (**a**) forces; (**b**) moments.

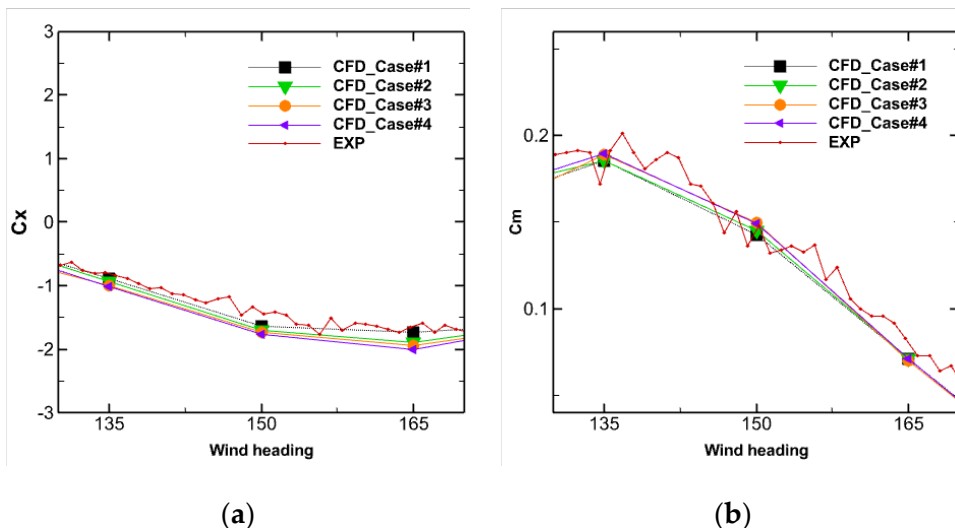

**Figure 8.** Comparison of aerodynamic coefficients on the FPSO for 4 cases. (**a**) forces; (**b**) moments.

*3.2. Numerical Verification*

The goal of the numerical verification procedure is to quantify the numerical uncertainty and is a purely numerical exercise. The MARIN V&V tool [26,27,33,34] has been used in this study. Numerical simulations were carried out for four vessels with four meshes with headings from 0° to 180°, at intervals of 15°. To obtain the numerical uncertainty, the following set of equations is solved:

$$\delta_{RE} = \phi_i - \phi_o = \alpha h_i^p$$

$$\delta_1 = \phi_i - \phi_o = \alpha h_i$$

$$\delta_2 = \phi_i - \phi_o = \alpha h_i^2$$

$$\delta_{12} = \phi_i - \phi_o = \alpha_1 h_i + \alpha_2 h_i^2, \tag{4}$$

where $\phi_i$ is the coefficient of interest obtained on grid $i$, $\phi_o$ is an estimate of the exact value $\phi_{exact}$, $\alpha$ is a constant, $h_i$ is the typical cell size of grid $i$ (using different initial mesh numbers), and $p$ is the observed order of grid convergence. These four equations were solved in the least-squares sense, with and without weights. The selected error estimator depends on the observed order of accuracy $p$ and the standard deviation of the fits. The value is given a larger weight on finer grids, where better results are expected. Once the error has been estimated, it is combined with a safety factor to yield the uncertainty estimator $U_\phi$ as follows:

$$\phi_i - U_\phi \leq \phi_{exact} \leq \phi_i + U_\phi, \tag{5}$$

where the comparison error $\phi_i$ is the difference between the simulated result $S_\phi$ and experimental measurement $D_\phi$

$$\phi_i = S_\phi - D_\phi. \tag{6}$$

$U_\phi$ is the validation uncertainty, which is a consequence of the numerical $U_S$ from grid refinement studies [35] and experimental $U_D$ from information reported in the literature, which unfortunately does not always provide the required values of $U_D$ and input parameter $U_I$ uncertainties, as follows:

$$U_\phi^2 = U_S^2 + U_D^2 + U_I^2. \tag{7}$$

Figure 9 shows the selection of the verification results of the FPSO and shuttle tanker for $C_x$ (180° heading), $C_y$ (90° heading), and $C_m$ (135° heading), in which the simulated

coefficients are indicated on the *y*-axis and the relative mesh (step) size on the *x*-axis. The relative step size can be calculated as:

$$\text{Relative step size} = \sqrt[3]{n_1/n_i}, \tag{8}$$

where $n_1$ is the total number of cells, and $n_i$ is the number of cells for mesh *i*. The finest mesh had a relative mesh size of 1, whereas the coarsest mesh had a relative mesh size of approximately 2. Ideally, it is desirable to use a wider range of relative mesh sizes. The mesh settings for the single vessels were chosen to intentionally limit the number of cells for simulations of a single vessel to ~25M cells to ensure that they were used in the same way for the side-by-side configuration cases.

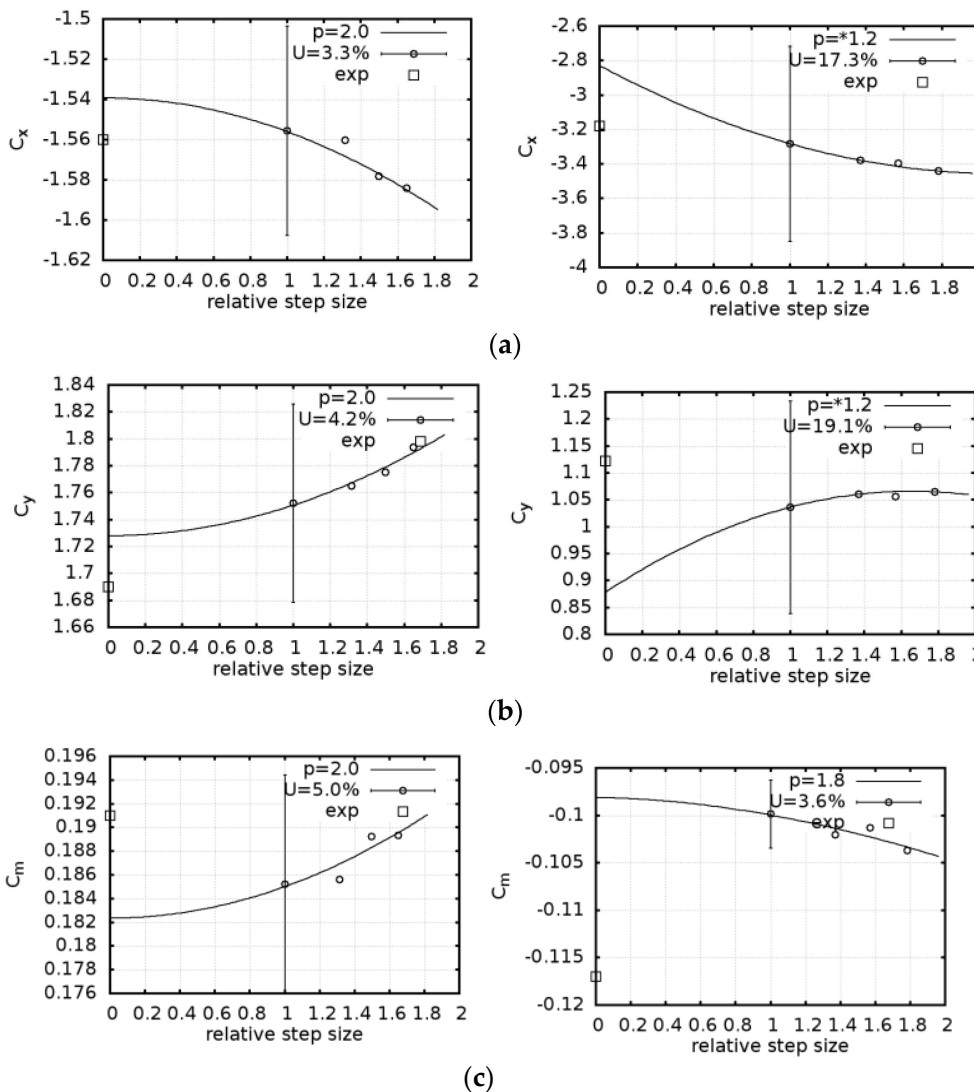

**Figure 9.** Verification results for FPSO (**left**) and shuttle tanker (**right**). (**a**) $C_x$ at 180° heading; (**b**) $C_y$ at 90° heading; (**c**) $C_m$ at 135° heading.

From all examples in Figure 9, as the simulated coefficients become monotonous with increasingly fine meshes, it seems that the numerical uncertainty (*U*) is low and the observed order of accuracy (*p*) value is close to 2, similar to that of the QUICK discretization scheme. This implies that the problem is not very sensitive to the mesh size used, and the numerical uncertainty is generally below 5%.

When the calculated coefficients are non-monotonically converging or diverging with grid refinement, a relatively large uncertainty is found. For diverging coefficients, the small

range in relative step size results in a large numerical uncertainty, not helped by a safety factor of 3, which is applied for non-converging coefficients. The fitted curve is not close to the order of the discretization method used. The large numerical uncertainty indicates that the results are very sensitive to a small change in the numerical mesh. This can, for instance, be due to the change in the separation point on the hull when the mesh is slightly changed. For these headings, it is important to have control over the numerical mesh to ensure that the numerical uncertainty is low. Modeling errors can be quantified only when the uncertainty is low.

### 3.3. Validation

After quantifying the numerical uncertainty, the results are compared to the wind tunnel experiments. For each vessel, the CFD results for the finest mesh are compared with the experiments in Figures 10–13, in which the uncertainty bars in the figures are obtained from the numerical uncertainty results. Even though the repeated uncertainty of the wind tunnel tests has been quantified, the wind tunnel tests do not have an uncertainty bar because the magnitudes of the other uncertainties cannot be quantified.

Overall, the CFD results were in close agreement with the wind tunnel experiments. For the FPSO, the largest deviation is found for $C_x$ at 105°, but owing to the large numerical uncertainty, this is not considered a significant deviation. Similar conclusions hold for the membrane-type LNGC, even though the uncertainty bars are very large. Each of the four simulated coefficients closely matches the experimental data; however, owing to the non-monotonic convergence or divergence, the uncertainty considered becomes large.

The most challenging vessel is the moss-type LNGC. There is still good agreement between the experiments and CFD, but the numerical uncertainties are larger for this vessel compared with the other vessels. As mentioned in Section 3.2, the separation point on the hemispherical tanks is very sensitive to the mesh size and has a large influence on the total drag of the vessel. Therefore, it was difficult to determine if the results were significant. Nevertheless, similar to the membrane-type LNGC, each of the four simulated coefficients per heading showed a very satisfactory agreement with the wind tunnel data within 10%.

As described above, it can be seen that good agreement between the simulated coefficients and wind tunnel data is represented in the case of single vessels. However, it is observed that some cases with small coefficients (not relevant angles) have large numerical uncertainties. In particular, the numerical uncertainty for moss-type LNGC is unrealistic because of the difficulty in capturing very complicated flows around hemispherical tanks. These large uncertainties may not make it easy to deduce meaningful conclusions about the deviations between CFD and the experiments.

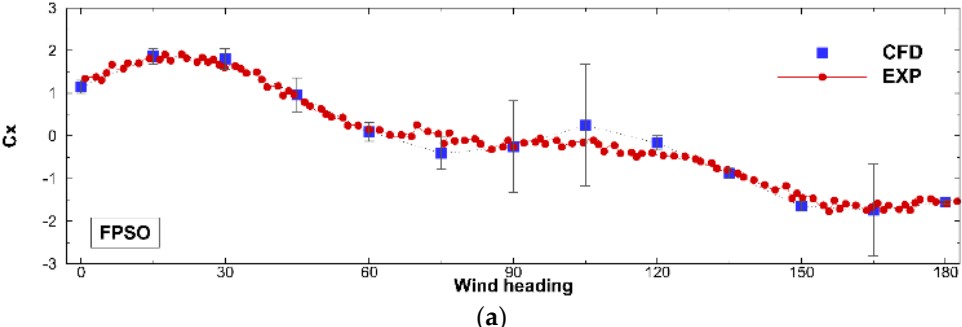

(a)

**Figure 10.** *Cont.*

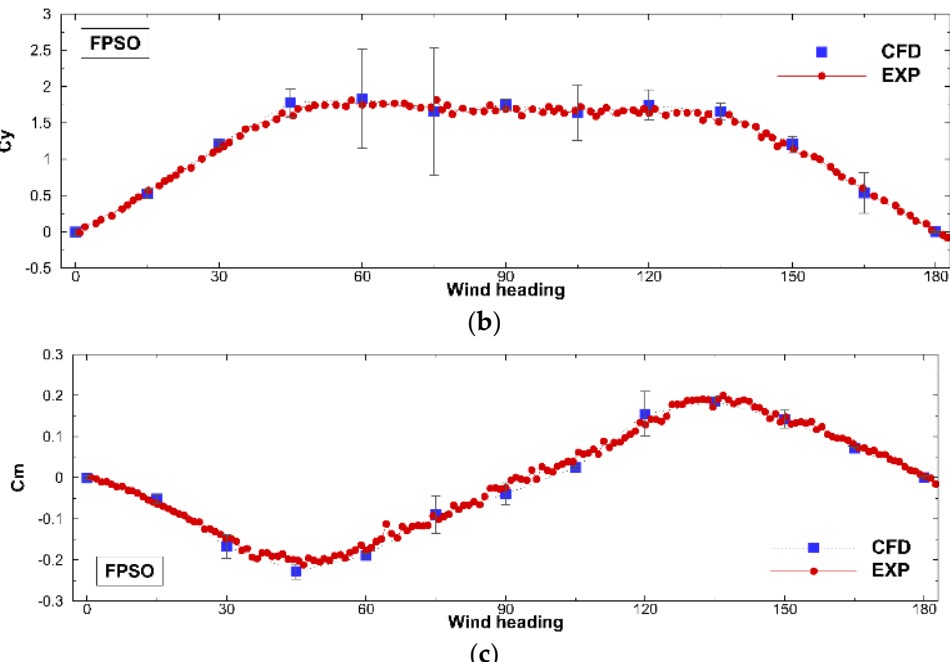

**Figure 10.** Comparison of aerodynamic coefficients for FPSO. (**a**) $C_x$; (**b**) $C_y$; (**c**) $C_m$.

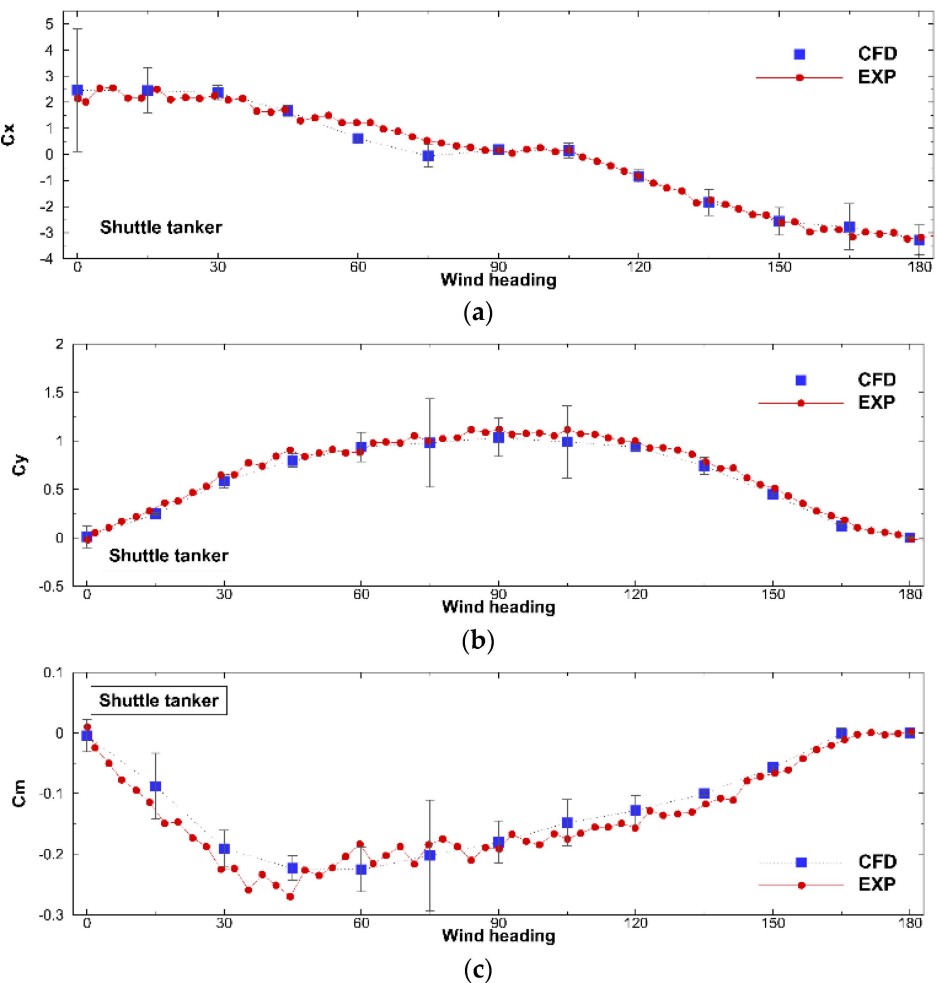

**Figure 11.** Comparison of aerodynamic coefficients for the shuttle tanker. (**a**) $C_x$; (**b**) $C_y$; (**c**) $C_m$.

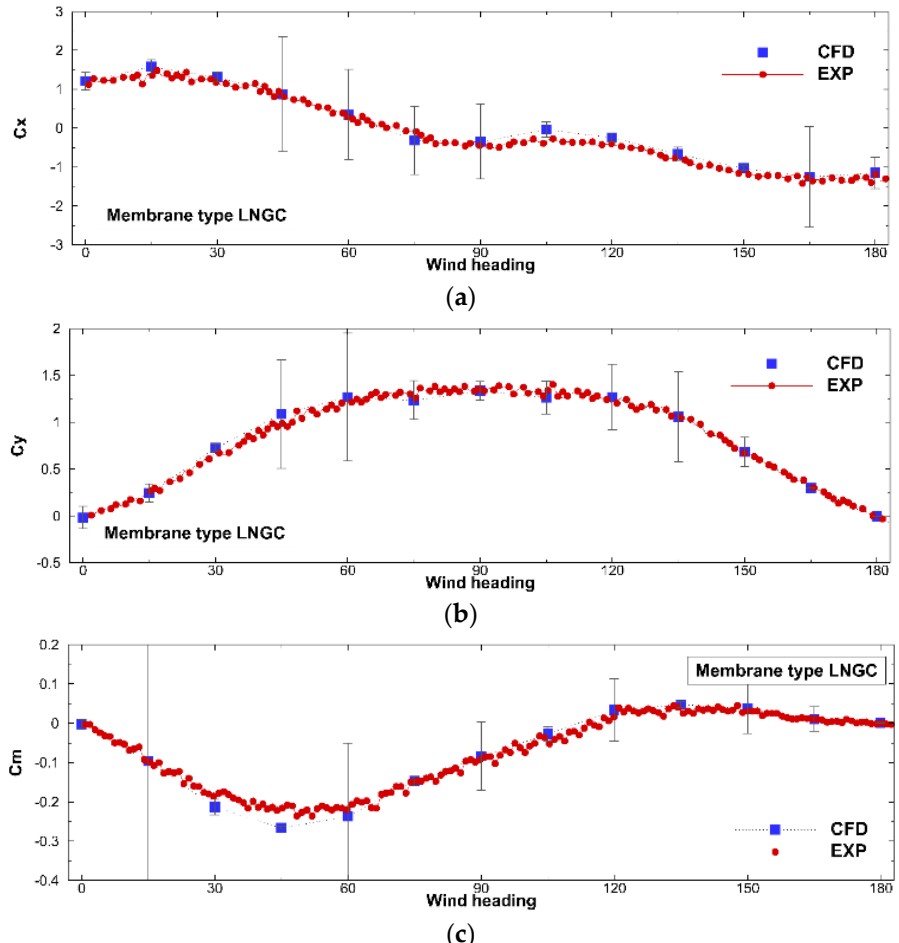

**Figure 12.** Comparison of aerodynamic coefficients for the membrane-type LNGC. (**a**) $C_x$; (**b**) $C_y$; (**c**) $C_m$.

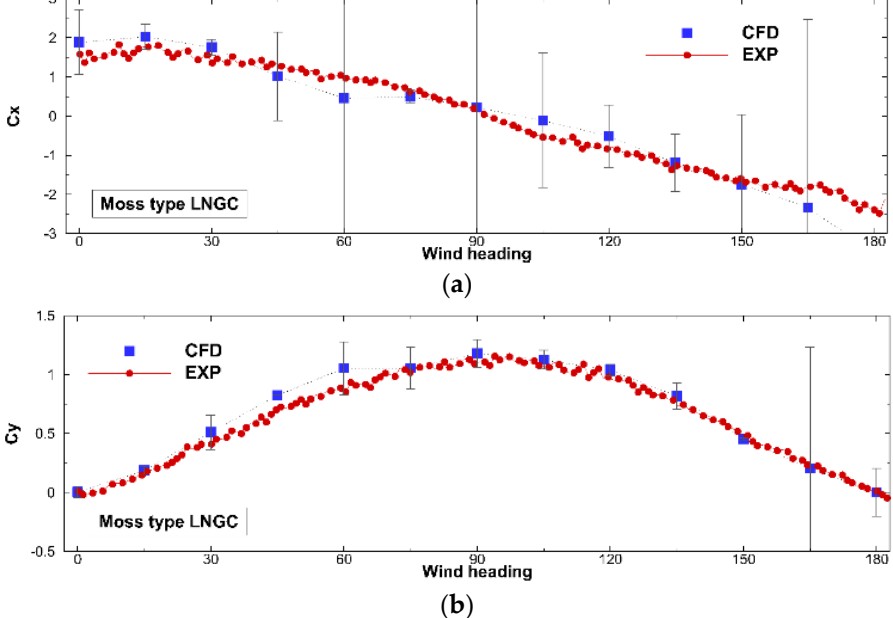

**Figure 13.** *Cont.*

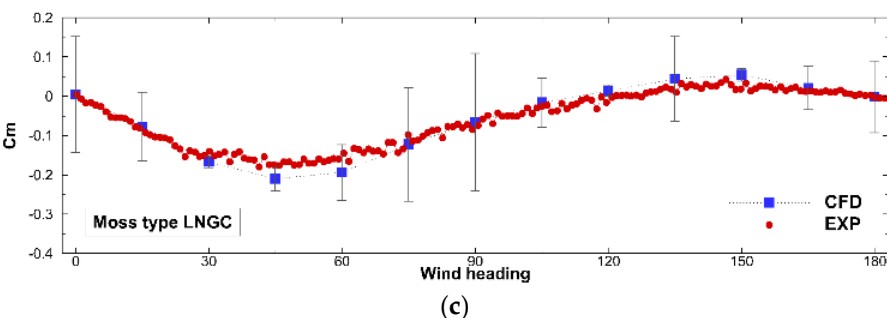

(**c**)

**Figure 13.** Comparison of aerodynamic coefficients for the moss-type LNGC. (**a**) $C_x$; (**b**) $C_y$; (**c**) $C_m$.

## 4. Side-by-Side Offloading Operation

In addition to measurements of single vessels, a tandem-offloading case and a side-by-side configuration were tested within the OO2 JIP [31]. CFD simulations of the tandem-offloading case were carried out previously [36]. Based on the single-vessel simulations presented in the previous section, the side-by-side configuration of the FPSO and shuttle tanker is the main focus.

### 4.1. Numerical Setup

The numerical setup for the side-by-side configuration was very similar to that for the single-vessel simulations. A cylindrical domain, with a radius of $6L_{ref}$ for the largest vessel (FPSO in this case) and a height of 2.25 m, is used. Because the size of the wake zone is more likely to be larger than that in the single-vessel case, owing to the interaction between two vessels, the domain is made slightly wider. The model geometry for the CFD simulations is shown in Figure 14. It can be observed that the FPSO is shielded by a shuttle tanker when the heading is between 0° and 180°, and the shuttle tanker is shielded by the FPSO between 180° and 360°.

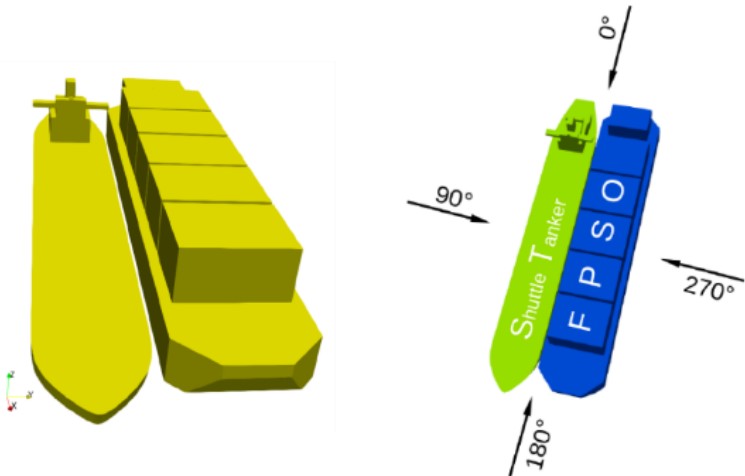

**Figure 14.** Configuration of FPSO and shuttle tanker and incidence angles for the CFD simulations.

Four mesh systems of different sizes were generated on the vessel surfaces with identical settings used for the simulations of a single vessel. Additionally, the mesh refinement is performed in the gap between the two vessels. When the vessels were spaced by 4 m, four mesh systems with different sizes were generated to perform the numerical verification, whereas for spacings of 10 m and 30 m, only the second finest mesh was used. The mesh system used is illustrated in Figure 15, and the number of cells is summarized in Table 3.

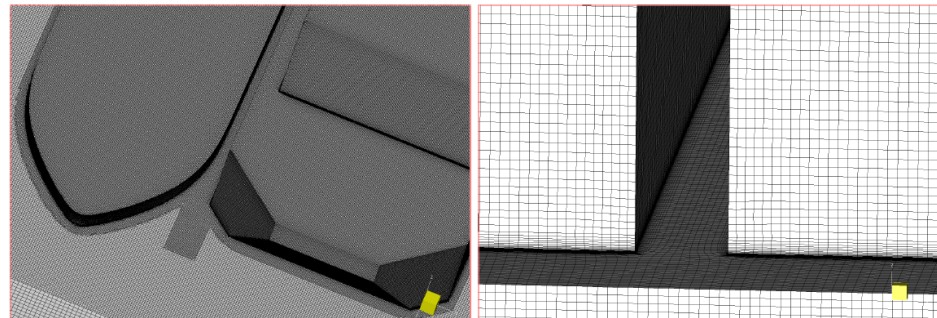

**Figure 15.** Finest grid system for the side-by-side case; (**left**) bow section including additional box refinement between the two vessels and (**right**) superstructure.

**Table 3.** Meshes used for the side-by-side simulations.

| Gaps (m) | Number of Cells | | | |
|---|---|---|---|---|
| | Case#1 | Case#2 | Case#3 | Case#4 |
| 4 | 85.6M | 36.6M | 24.8M | 17.6M |
| 10 | - | 37.9M | - | - |
| 30 | - | 44.0M | - | - |

## *4.2. Verification Study*

In the case of a gap of 4 m, as mentioned above, simulations were performed on the mesh systems with four different sizes, with 45° intervals in the range of 0°–315°. The aerodynamic forces and moments on both the FPSO and shuttle tanker were extracted through the simulation. The simulation results of the numerical verification for $C_x$ at 180° and $C_y$ at 90° are presented in Figure 16. Similar to the single-vessel cases, the range of the relative step size is not sufficient owing to the computational limitations. Assuming the coarsest mesh of 17.6M, it seems desirable that the relative step size for Case #4 is 1126M cells, which is an unrealistic number of meshes.

In general, a small relative mesh size seems to have a significant influence on the computed numerical uncertainty. For example, the shuttle tanker at 180° provides a numerical uncertainty of 89.5% for $C_x$ in Figure 16. Owing to the non-monotonic convergence, a large numerical uncertainty could be obtained, and the uncertainty bar ranges from −0.22 to −4.2. However, the simulated coefficients for the four mesh cases vary in the range of −2.2 to −2.4, which implies that the relative errors occur within 10% for each individual mesh compared with the experiment that results in a value of −2.5. For these complex cases, it can be argued that the numerical uncertainty tool does not provide a realistic representation of the uncertainty; instead, the uncertainty should be computed using an alternative method.

Thus, the verification procedure carried out here should be handled with care. Owing to the small range of relative mesh sizes, the small number of cases, and the imperfect geometric similarity of the mesh, large uncertainties can be present. As a result, the actual error of the calculations compared with the wind tunnel experiments could not be clearly determined.

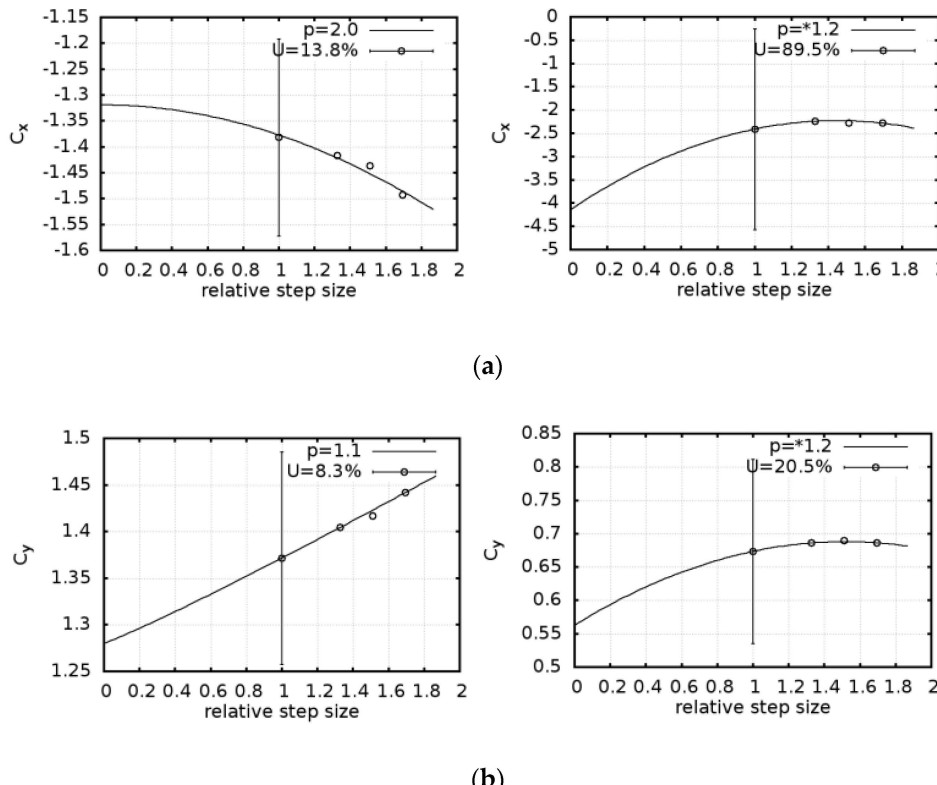

(a)

(b)

**Figure 16.** Verification results for the FPSO (**left**) and shuttle tanker (**right**). (**a**) $C_x$ at 180° heading; (**b**) $C_y$ at 90° heading.

*4.3. Validation*

The simulated coefficients for the FPSO and shuttle tanker with a 4 m gap size are compared with the wind tunnel data for the FPSO and shuttle tanker in Figures 17 and 18, respectively. The CFD simulations were conducted for the second finest mesh, consisting of 36.6M cells, at steps of 15° with a heading from 0° to 360°. The CFD results are presented, including the numerical uncertainties obtained from the numerical verification in the previous section.

The FPSO is shielded by the shuttle tanker between 0° and 180°. The volume and projected area of the FPSO are much larger than those of the shuttle tanker, shielding effects on the tanker are therefore very limited. All simulated coefficients show good agreement with the experimental data, and it can be seen that the relative errors in the experiment for all headings are within 3% on average.

For the shuttle tanker, the CFD results agreed well with the experimental data. The shuttle tanker was shielded by the FPSO between 180° and 360°. The CFD simulation reproduced the experimental trend very similarly, but it tended to deviate somewhat at some headings. Although the maximum relative error of $C_m$ at 330° was approximately 12%, the average relative error was within 5%.

Figure 19 shows a comparison of the Q-criterion (measure of the vorticity, iso-surface at $Q = 10^5$) for the side-by-side configuration (bottom) and single-vessel cases (two tops) observing the region disturbed by the shuttle tanker at 45°. It can be clearly understood that the shuttle tanker with few superstructures has a relatively small shielding effect on the FPSO, but the flow separation and resulting vortices from the accommodation deck at the stern influence the FPSO. In addition, it can be seen that the size of the iso-volume representing the number of vortices around the accommodation part is remarkably different owing to the interaction with the FPSO.

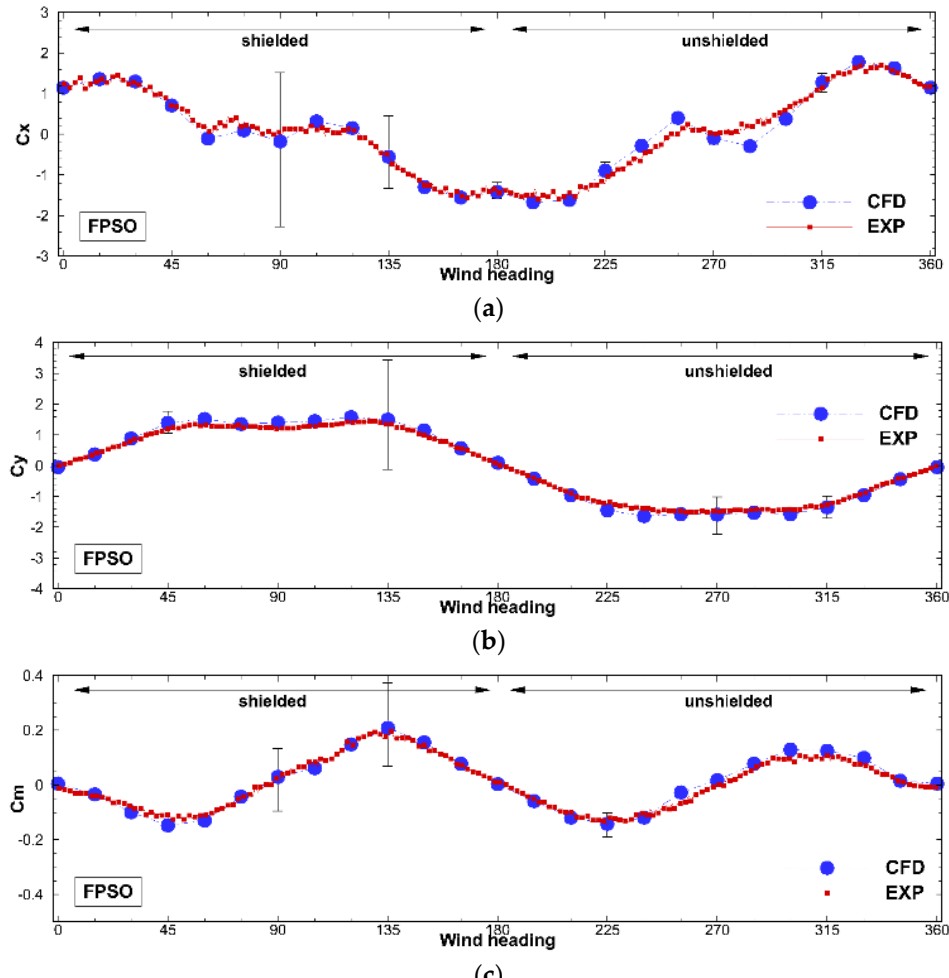

(a)

(b)

(c)

**Figure 17.** Comparison of aerodynamic coefficients for the FPSO in the side-by-side configuration (4 m gap). (**a**) $C_x$; (**b**) $C_y$; (**c**) $C_m$.

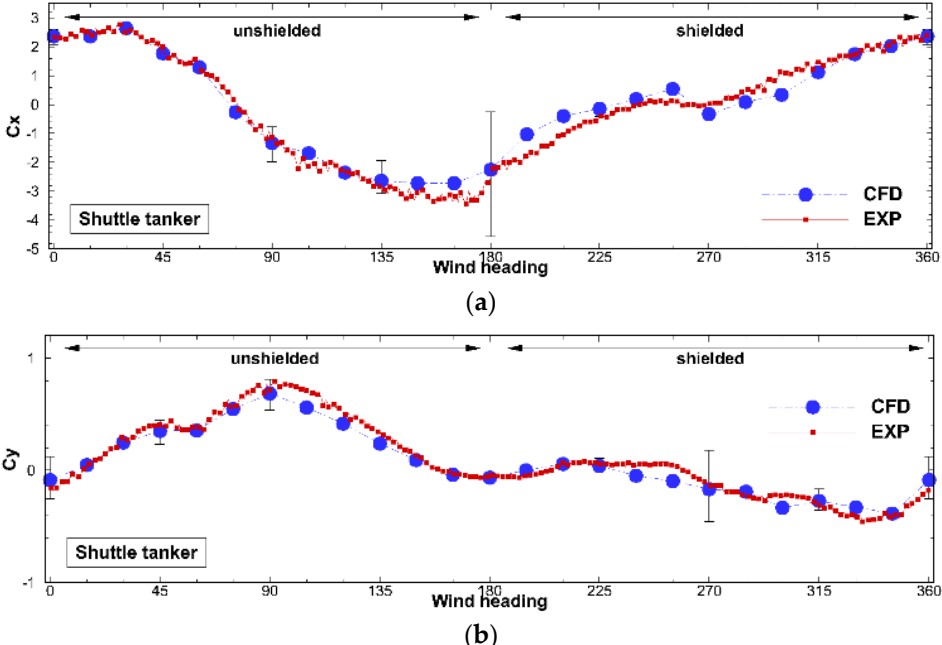

(a)

(b)

**Figure 18.** *Cont.*

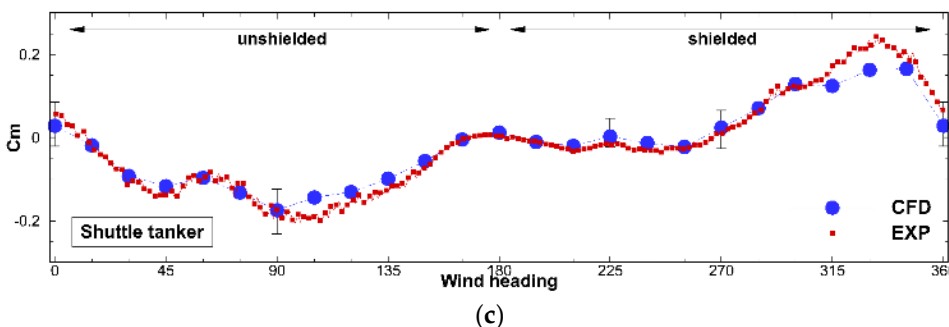

(**c**)

**Figure 18.** Comparison of aerodynamic coefficients for the shuttle tanker in the side-by-side configuration (4 m gap). (**a**) $C_x$; (**b**) $C_y$; (**c**) $C_m$.

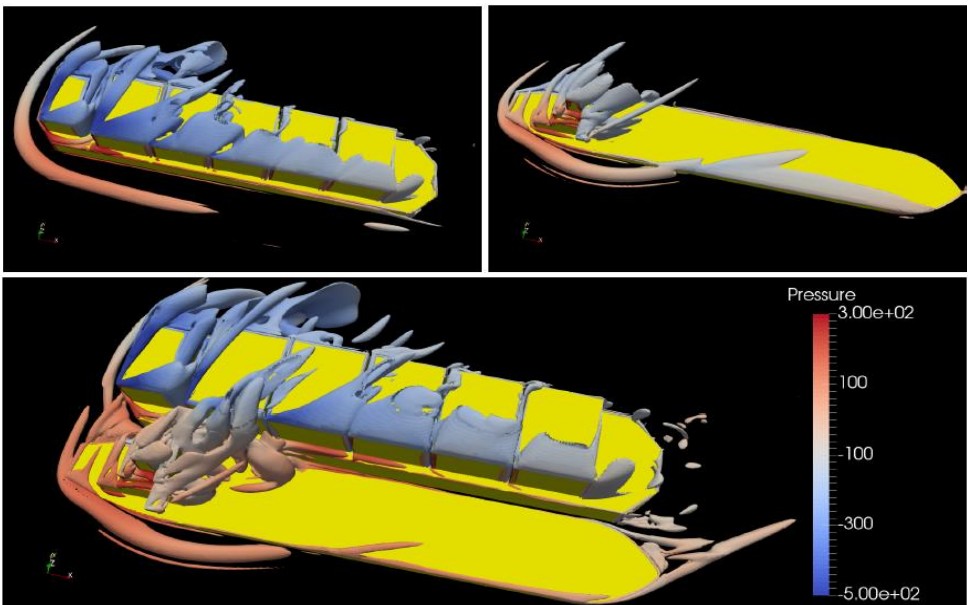

**Figure 19.** Comparison of Q-criterion between single vessels (**two tops**) and side-by-side configuration (**bottom**).

### 4.4. Gap Effects between Two Vessels

With good agreement between the experiments and CFD, it is possible to investigate variations in the side-by-side configuration. Additional simulations were performed with gap sizes of 10 m and 30 m for the arrangement between the two vessels.

The simulated coefficients according to the headings for each vessel with different gaps are presented in Figures 20 and 21, compared with the experimental data of single-vessel cases. For the FPSO in Figure 20, the presence of the shuttle tanker has only a limited influence on the FPSO, and the coefficients do not appear to differ significantly from those of a single vessel for all gap sizes. As shown in Figure 21, on the other hand, the shuttle tanker is much more heavily shielded by the large FPSO, and it is observed that there are significant differences in the coefficients for the side-by-side configuration compared to those of the single-vessel case. In particular, $C_y$ tends to decrease significantly compared with that of a single vessel in the entire range of headings, which is due to the presence of the FPSO causing large-scale flow separation generated in the wake zone of the tank, suppressed in the unshielded range (0°–180°) and owing to the shielding effect of FPSO in the shielded range (180°–360°). For the variation of gap sizes, it seems to be less sensitive to the coefficients; however, the results at 30 m, the largest gap, move slightly closer to those of the single vessel. In other words, it is understood that the shuttle tank gains a benefit in

wind load owing to the shielding effect during a side-by-side offloading, and the larger the gap distance between the two vessels the lower the shielding effect.

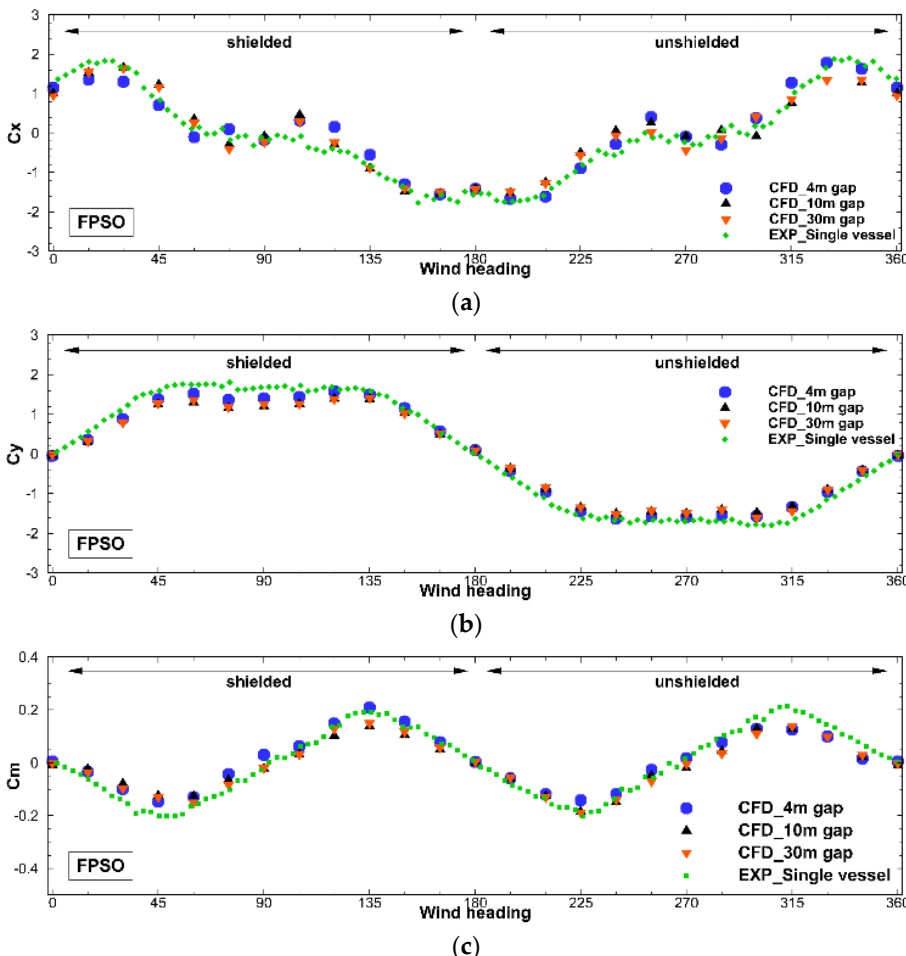

**Figure 20.** Comparison between single-vessel case and side-by-side case with different gaps for the FPSO. (**a**) $C_x$; (**b**) $C_y$; (**c**) $C_m$.

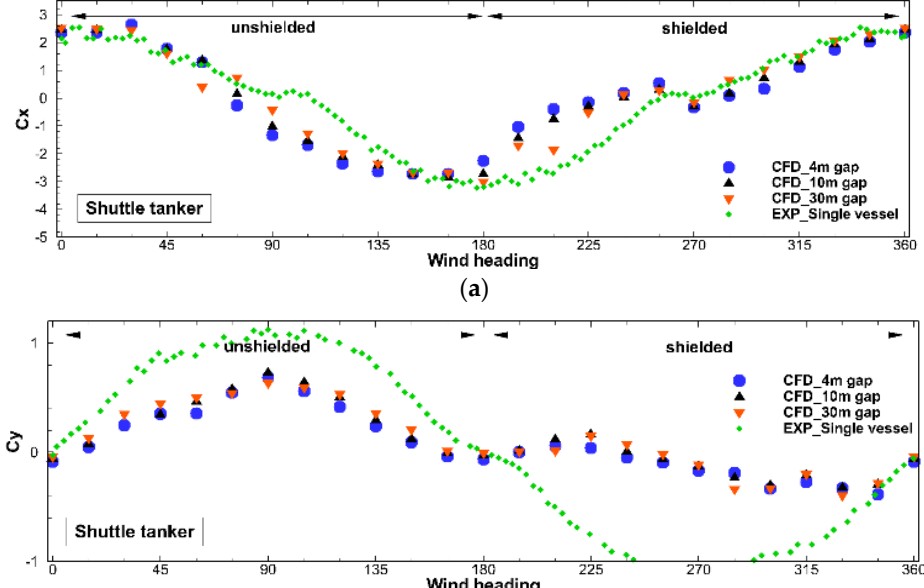

**Figure 21.** *Cont.*

**(b)**

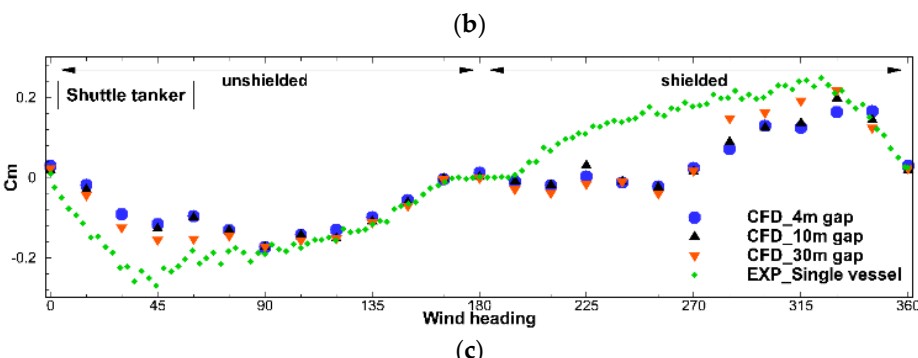

**(c)**

**Figure 21.** Comparison between single-vessel case and side-by-side case with different gaps for the shuttle tanker. (**a**) $C_x$; (**b**) $C_y$; (**c**) $C_m$.

## 5. Conclusions

In this study, CFD simulations for estimating wind loads were performed for four typical offshore vessels and for an FPSO and shuttle tanker in a side-by-side configuration. The numerical uncertainty for all vessels was investigated using four different meshes according to the guidelines of the V&V studies [26]. The simulated aerodynamic coefficients (forces and moments) were compared with the wind tunnel experiments provided by OO2 JIP [31]. The conclusions obtained from this study are summarized as follows:

- From the CFD results of the single-vessel cases, good agreement with the experimental data was found for all vessels. The most challenging case was the moss-type LNGC with hemispherical tanks. Despite the large numerical uncertainty, the numerical modeling error was small and a good agreement with the wind tunnel tests was obtained;

- In the case of a side-by-side configuration with a combination of an FPSO and shuttle tanker at a gap of 4 m, larger numerical uncertainties were found, especially when the vessel was shielded. Despite the relatively large numerical uncertainty, agreement with wind tunnel data was good, in which all coefficients were within 3–5% of the experiments on average.

- To identify the gap effects between two vessels in a side-by-side arrangement, CFD simulations were performed for two cases, with gaps of 10 m and 30 m. While FPSO had only a limited effect because of the existence of the shuttle tanker, the shuttle tanker showed reduced aerodynamic force and moment in almost all heading ranges compared with a single vessel owing to the strong shielding effect of the larger FPSO. In addition, as the gap between the two vessels increased, the shielding effect gradually decreased and eventually approached the wind load acting on a single vessel.

**Author Contributions:** Conceptualization, P.S., A.K. and J.-C.P.; writing—original draft preparation, J.-H.Y.; writing—review and editing, P.S. and J.-C.P.; software, J.-H.Y.; validation, J.-H.Y.; visualization, J.-H.Y.; supervision, P.S. and J.-C.P.; project administration, J.-C.P. All authors have read and agreed to the published version of the manuscript.

**Funding:** This work was partly supported by the Technology Innovation Program (20008690, Optimal hull cleaning and propeller polishing scheduling for minimal ship operating cost using operating performance analysis) funded by the Ministry of Trade, Industry & Energy (MOTIE, Korea). This research was also partly funded by the Dutch Ministry of Economic Affairs.

**Institutional Review Board Statement:** Not applicable.

**Informed Consent Statement:** Not applicable.

**Data Availability Statement:** Request to the corresponding author of this article.

**Acknowledgments:** Note that this paper has been added and supplemented to the manuscript presented in ISOPE2018 [37].

**Conflicts of Interest:** The authors declare no conflict of interest.

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
