# Peer review of "CFD Prediction of Wind Loads on FPSO and Shuttle Tankers during Side-by-Side Offloading"

_jmse, doi:10.3390/jmse10050654_

Round 1

Reviewer 1 Report

The authors performed a CFD study on the wind loads in 4 distinct vessels and then on side-by-side configurations. The paper is very well written with the procedures and main points all very well described and scientifically sound.

The conclusions are supported by the results, which are then clearly presented and commented. Furthermore, error and uncertainties are well treated, although with some limitations in respect to the experimental results. The numerical results are also (well) compared with wind-tunnel experiments.

The study on varying gap distance in the end of the manuscript provide interesting conclusions that are valuable for researchers/engineers dealing with this kind of problem.

Author Response

Thank you very much for your kind comments.

Reviewer 2 Report

Very interesting paper dealing with very usefull investigation of the CFD prediction of wind loads on different ships and for an FPSO and shuttle tanker in a side-by-side configuration. The paper is comprehensible and very well organized and written.

Author Response

Thank you very much for your insightful comments.

Reviewer 3 Report

In a word, the paper is well written  but there are some issues required to be addressed before being accepted.

1.In Section 2.1, it is necessary to illustrate how to model the wind flow in CFD code, is it compressible or incompressible flow? please add some equations to describe the model.
2. The comparison of Figure 6(a) and (b) is too hard to evaluate if the independence of mesh are obtained or not, so it is recommended to add all results of 4 meshes from coarsest mesh to finest mesh in one figure.
3. V&V is a very important aspect in this paper to make the CFD simulated results convincing. In the introduction part, the authors have given some introductions about V&V but there are many other new approaches or studies for V&V analysis missing, so it is recommended to introduce grid convergence index (GCI) or uncertainty analyses proposed by following reference for studying sensibility of the meshes. eg. A procedure for the estimation of the numerical uncertainty of CFD calculations based on grid refinement studies(JCP).  A CFD Approach for Numerical Assessment of Hydrodynamic Coefficients of an Inclined Prism near the Sea Bottom(OE).

Author Response

Thank you for reviewer's comments. Please Note that Authors’ responses are written in the enclosed.
